# An Investigation of Bacterial Pathogens Associated with Diseased Nile Tilapia in Small-Scale Cage Culture Farms on Lake Kariba, Siavonga, Zambia

**Mazuba Siamujompa** [1,2], **Kunda Ndashe** [1,*], **Frederick Chitonga Zulu** [1], **Chanda Chitala** [1,3], **Mwansa M. Songe** [3], **Katendi Changula** [1], **Ladslav Moonga** [1], **Emmanuel Shamulai Kabwali** [1], **Stephen Reichley** [4,5] **and Bernard Mudenda Hang'ombe** [1]

[1] Department of Paraclinical Studies, School of Veterinary Medicine, The University of Zambia, Lusaka P.O. Box 32379, Zambia; msiamujompa@gmail.com (M.S.); frederickchito@gmail.com (F.C.Z.); cchitala@yahoo.co.uk (C.C.); katendi.changula@gmail.com (K.C.); ladslav.moonga@unza.zm (L.M.); emmanuelkabwali@yahoo.com (E.S.K.); bhangombe@unza.zm (B.M.H.)

[2] Department of Zoology and Aquatic Sciences, Faculty of Natural Resources, Copperbelt University, Kitwe P.O. Box 21692, Zambia

[3] Central Veterinary Research Institute, Department of Veterinary Ministry of Fisheries and Livestock, Lusaka P.O. Box 33980, Zambia; drsonge@yahoo.com

[4] Global Center for Aquatic Health and Food Security, Mississippi State University, Starkville, MS 39762, USA; stephen.reichley@msstate.edu

[5] Department of Pathobiology and Population Medicine, College of Veterinary Medicine, Mississippi State University, Starkville, MS 39762, USA

* Correspondence: ndashe.kunda@gmail.com; Tel.: +260-967-903-076

**Abstract:** This study investigated disease outbreaks in farmed *Oreochromis niloticus* (Nile tilapia) in Siavonga among small-scale cage culture farms on Lake Kariba in order to establish bacterial etiological agents associated with fish mortality and to determine their antibiotic susceptibility. A total of 300 fish samples from 11 farms were aseptically collected and bacteria were isolated from the kidney, liver, brain, and spleen. The isolates were identified using their morphological characteristics and conventional biochemical tests. The antibiotic susceptibility of selected bacteria was determined by the Kirby–Bauer disc diffusion method. The following well-known fish pathogens were identified at a prevalence of *Aeromonas* spp. (13%), *Pseudomonas* spp. (10.3%), *Micrococcus* spp. (9.7%), *Klebsiella* spp. (8.7%), *Lactococcus* spp. (7.3%), *Streptococcus* spp. (7.0%), and *Acinetobacter* spp. (7.0%). All the isolates tested were susceptible to doxycycline, and complete resistance to ciprofloxacin, co-trimoxazole, and cephalothin was recorded in the *Bacillus* spp. The observed resistance could be attributed to bacteria from terrestrial sources as fish farmers do not administer antibiotics to fish. To our knowledge, this is the first study to establish the occurrence of several bacterial species infecting tilapia in Zambia and the first to determine the antibiotic susceptibility of fish bacteria among small-scale farms on Lake Kariba. The current study provides baseline information for future reference and fish disease management on Lake Kariba and in Zambia.

**Keywords:** Nile tilapia; fish bacteria; antibiotic resistance; small-scale; Zambia

**Key Contribution:** This study represents a pioneering endeavor to elucidate the bacterial pathogens that impact farmed Nile tilapia within the burgeoning community of small-scale producers situated along Lake Kariba in Zambia.

## 1. Introduction

Zambia has recorded significant growth in the aquaculture sub-sector, recording 400% growth in the past decade [1]. Significant investment in the aquaculture industry has seen the establishment of fish farms and other allied industries throughout the country [2].



Furthermore, improved production technologies and systems have contributed to improved yields per farm.

Globally, the growth of the aquaculture industry has been accompanied with corresponding increases in the incidence of disease outbreaks [3]. Diseases are a major cause of economic loss in the aquaculture industry [4]. Bacterial diseases are the most frequent and major cause of mass death in fish and are the main pathogens of cultured tropical freshwater fish [5,6]. Many bacteria are commensal to fish and other aquatic organisms or live free in the environment, while others are opportunistic [6]. Opportunistic bacterial infections can occur when the fish are immunosuppressed by the effects of different stressors [7]. Some of the pathogenic bacteria that can cause infections in tilapia fish include *Pseudomonas* spp., *Aeromonas* spp., *Vibrio* spp., *Streptococcus* spp., *Micrococcus* spp., *Enterococcus* spp., *Lactococcus* spp., *Edwardsiella* spp., and *Flavobacterium* spp. [8–12].

Baseline information on the antibiotic susceptibility of bacteria in an epidemiological zone or region is vital for the proper management of the diseases they cause [13]. The development of antibiotic resistance by bacteria in an aquatic environment can occur through the potential transmission of resistant bacteria between terrestrial and aquatic environments, the abuse of antibiotics in bacterial infections, and other factors [13]. There is particular attention being paid to the use of antibiotics in fish farming due to the possibility of antibiotic residues contaminating the aquatic environment, animals, and plants, and the terrestrial animals that consume the water. Ndashe et al. (2023) reported that fish farmers in Zambia do not use antibiotics in the treatment of diseases [14]. However, with the growth of the sector in Zambia, the use of antibiotics in the treatment of bacterial diseases will become inevitable.

Several studies have reported the diagnosis and detection of pathogens in fish disease outbreaks on Lake Kariba; however, they have been limited to commercial farms [15–17]. However, there has been no study to our knowledge on the small-scale fish farms situated on Lake Kariba. This study, therefore, investigated bacterial disease outbreaks among small-scale fish farms in the Siavonga district, Zambia. Pathogens were isolated from farmed Nile tilapia (*Oreochromis niloticus*) and identified, followed by the evaluation of their antimicrobial susceptibility.

## 2. Materials and Methods

### 2.1. Study Area

The study was conducted in the Siavonga district (16.5323° S, 28.7111° E) in the Southern Province of Zambia (Figure 1). The study area was selected based on the relatively large number of small-scale cage fish farms. For the purposes of this study, a small-scale fish farm was defined as any production facility with an annual production of less than 100 tons using 6 by 6 m cages for rearing fish.

### 2.2. Study Design and Sampling

This study was cross-sectional and was conducted between October 2021 and January 2022 to identify the bacterial pathogens from diseased farmed tilapia. The water temperature during the study period was 27 to 30.2 °C. Moribund fish (190 to 372 g) were collected from all small-scale fish farms that reported high mortality during the study period. The moribund fish swimming at the surface were collected from affected cages that reported high mortality within 5 days preceding the farm visit. The fish were humanely euthanized using clove powder at 250 mg/L of water.

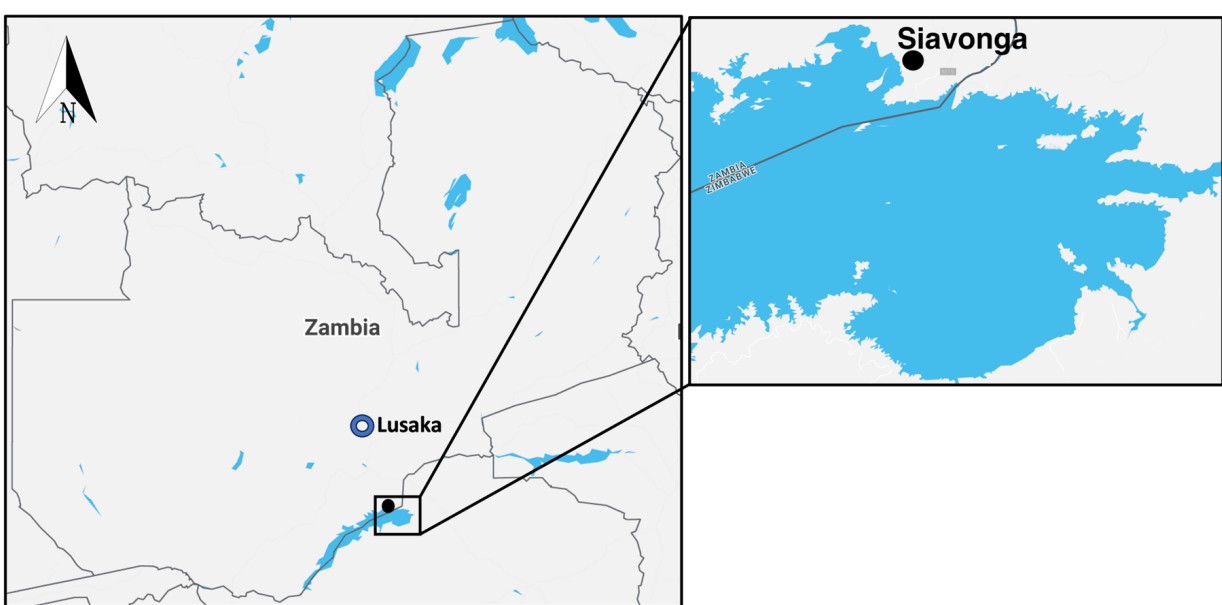

**Figure 1.** Map of Zambia with an insert of the northern region of Lake Kariba where Siavonga, the study area, is located.

### 2.3. Assessment of Morphological/Clinical Pathological Symptoms

Infected fish were thoroughly examined and gross lesions such as pale gills, exophthalmia, corneal opacity, abdominal distension, shallow ulcers, skin and fin haemorrhages, and fin erosion were recorded.

### 2.4. Necropsy and Bacterial Inoculation

Necropsy examinations were performed onsite within 30 min of euthanasia. An incision was made on the left side of the trunk around the abdomen, removing the flank and exposing all abdominal viscera. From each moribund fish, the brain, liver, and kidney were sampled using sterile disposable inoculation loops and streaked on MacConkey agar (HiMedia, Mumbai, India), nutrient agar (HiMedia, India), and blood agar (HiMedia, India) using a strictly aseptic technique.

The inoculated petri dishes were stored at room temperature (approximately 28 °C) and transported to the bacteriology laboratory, School of Veterinary Medicine, the University of Zambia within 24 h of sampling.

### 2.5. Biochemical Characterization

At the bacteriology laboratory, pure cultures were obtained by carrying out subculturing procedures and subjecting them to a second round of incubation at room temperature for another 48 h—ensuring the isolation of uncontaminated bacterial strains. The isolates were identified by determining their colony morphology, which included their shape, colour, pigmentation, haemolytic activity, size, edges, and elevation; afterwards, the isolates were grouped accordingly. Two to three representative isolates from each group were subjected to Gram-staining. Conventional biochemical tests and the API system were then used to characterize the bacteria (Table 1). Briefly, a loopful of bacteria was aseptically added to 5 mL of phenol red broth containing 1% sugar and incubated at 37 °C for 24 h to test for the fermentation of different sugars. Sulphur reduction, indole production, and motility tests were determined using sulphide indole motility (SIM) media again by adding a loopful of bacteria and incubating at 37 °C for 24 h.

**Table 1.** Biochemical characteristics of bacterial isolates isolated from tilapia (*Oreochromis niloticus*) and water samples from the study area.

| Bacterial Isolates | GM | CM | H | Ox | Cat | Esc | Gal | Raf | Sal | Mal | Xyl | Man | Tre | Inu | Sor | Lac | Urea | Glu | Suc | Lac | G | H2S | Cit | Sul | Mot | Ind | MR |
|---|---|---|---|---|---|---|---|---|---|---|---|---|---|---|---|---|---|---|---|---|---|---|---|---|---|---|---|
| *Acinetobacter* | − | Rod | β | + | + | − | − | − | − | − | − | − | − | − | − | − | − | − | − | − | − | − | + | − | − | − | − |
| *Aeromonas* spp. | − | Rod | β | + | + | − | + | − | − | + | − | − | + | − | − | − | − | + | + | + | − | − | − | − | + | + | − |
| *Bacillus* | + | Rod | β | | + | − | − | − | − | − | − | − | − | − | − | − | − | − | − | − | − | − | + | − | + | − | − |
| *Citrobacter* spp. | − | Rod | | − | + | − | − | − | − | − | − | − | − | − | − | − | + | − | − | − | − | − | + | − | + | − | + |
| *Klebsiella* spp. | − | Rod | | | | − | + | + | − | + | + | + | + | − | + | + | − | + | + | + | + | + | + | + | + | + | − |
| *Lactococcus* spp. | + | Cocci | α | − | − | − | + | − | − | − | − | − | − | − | − | − | + | + | + | + | − | − | − | − | + | − | + |
| *Micrococcus* | + | Cocci | | + | + | − | − | − | − | − | − | − | − | − | − | − | − | − | − | − | − | − | − | − | − | − | − |
| *Pseudomonas* spp. | − | Rod | | + | + | − | − | − | − | − | − | − | − | − | − | − | − | − | − | − | − | − | + | − | + | − | − |
| *Staphylococcus* spp. | + | Cocci | β | − | + | − | + | − | − | + | + | + | + | − | + | − | − | + | + | + | + | − | − | − | − | + | − |
| *Streptococcus* spp. | + | Cocci | | − | − | − | − | − | − | + | − | − | − | − | − | − | − | + | + | + | − | − | − | − | + | − | + |

GM = Gram stain, CM = cell morphology, H = hemolysis, Cat = catalase, Ox = oxidase, Esc = esculin, Gal = galactose, Raf = rafnose, Sal = salicin, Mal = maltose monohydrate, Xyl = xylose, Man = mannitol, Tre = trehalose, Inu = inulin, Sor = sorbitol, Lac = lactose monohydrate, Urea = urease, Glu = glucose, Suc = sucrose, Lac = lactose, G = gas production, H2S = production of hydrogen sulphide, Cit = citrate, Sul = sulphur, Mot = motility, Ind = indole, and MR = methyl red.

### 2.6. Determination of the Antibiotic Susceptibility of the Isolates

The antibiotic susceptibility of the representative isolates was determined using the disc diffusion method on Mueller–Hinton agar (HiMedia) [18]. Six antibiotic discs (HiMedia) were employed for the study: Ciprofloxacin (30 μg/disc), doxycycline (30 μg/disc), cephalothin (30 μg/disc), co-trimoxazole (25 μg/disc), ampicillin (25 μg/disc), and oxytetracycline (30 μg/disc), as per the manufacturer's (HiMedia) instructions. Antibiotic susceptibility was analysed by measuring the diameter of the zone of inhibition (mm) and the results were interpreted as according to the CLSI methods [18]. The antimicrobial agents were selected as representatives of the different classes of antibiotics used in aquaculture practice.

### 2.7. Data Analysis

Data were entered into an Excel spreadsheet (Microsoft Excel 2010 version, Redmond, WA, USA) and then exported to DATA Tab™ (Styria, Austria)—a Web-App for statistical data analysis—where descriptive statistics (frequencies and proportions) were computed and presented using tables for categorical parameters. The summary tables and graphs were prepared in accordance with the objectives of the study.

## 3. Results

A total of 300 samples were collected from 11 farms (27 ± 15 samples per farm) over the duration of the study.

### 3.1. The Clinical and Necropsy Findings

The clinical signs observed in situ were lethargy and erratic swimming. On Necropsy, external lesions such as haemorrhages on the skin surface, corneal opacity, skin discoloration, ulcers on the trunk, missing scales, fin erosion, and petechial haemorrhages on the operculum were recorded (Figure 2). Internal lesions noted included pale gills, pale liver, enlarged spleen, and enlarged anterior and/or posterior kidney.

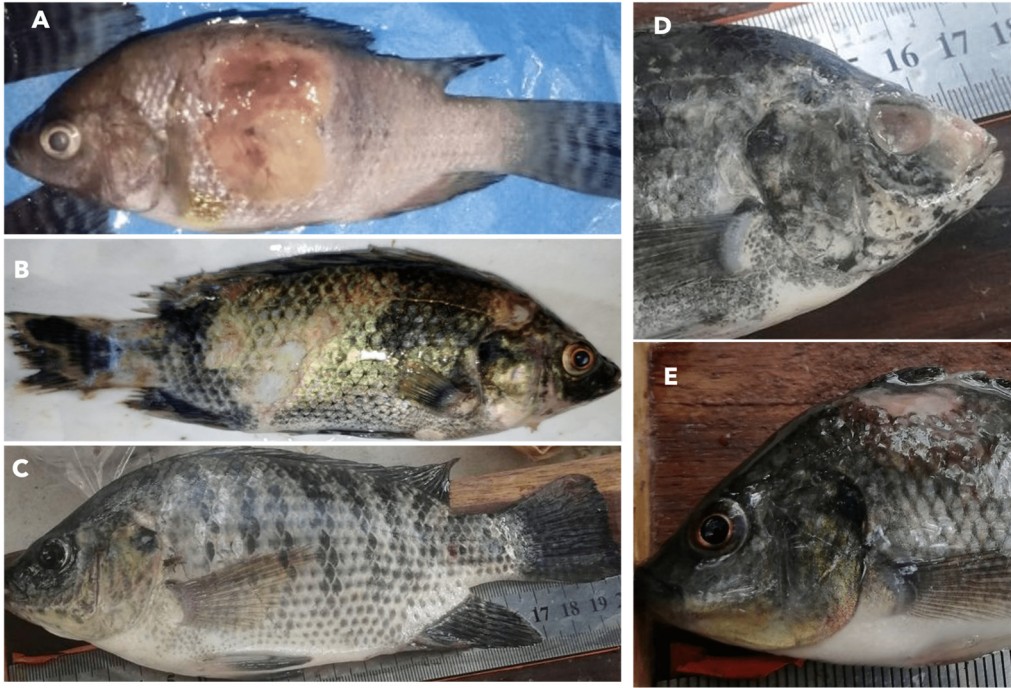

**Figure 2.** Macroscopic external lesions observed on diseased fish: (**A**) loss of scales and ulceration on the trunk; (**B**) Erosion of fins (dorsal, caudal, pectoral, and anal), loss of scales, and ulceration; (**C**) Abdominal distention; (**D**) Corneal opacity; (**E**) Deep ulceration on the head.

The external examination revealed a significant prevalence of lesions in the moribund fish, with skin discolouration (38%) and corneal opacity (29%) being particularly prominent (Figure 3).

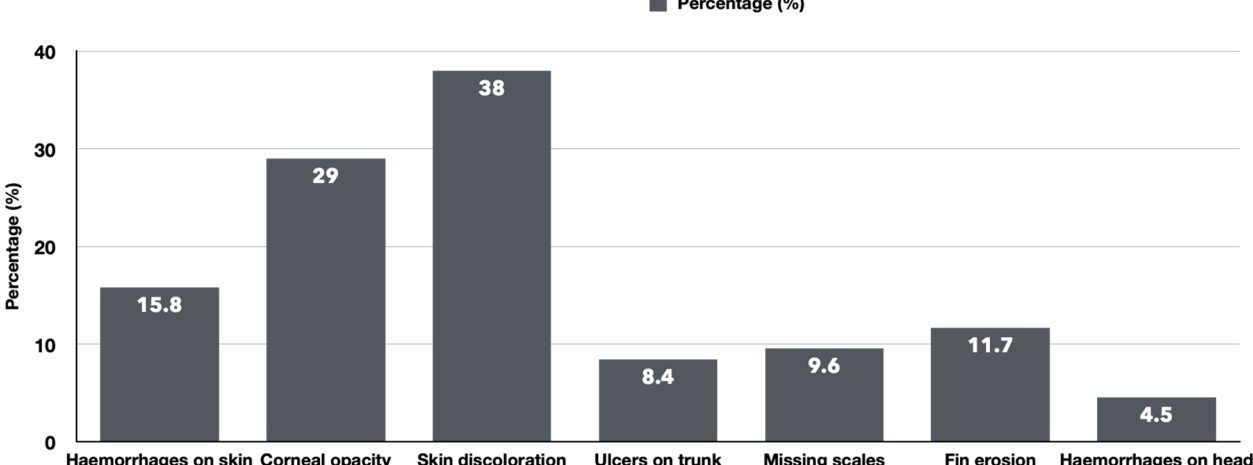

**Figure 3.** The percentage of fish with external pathological lesions recorded in the moribund fish.

During the internal examination conducted during necropsy, various lesions were detected among the viscera of the affected fish. Splenomegaly (22.6%), pallid livers (19.4%), anaemic gills (13.7%), haemorrhages on the brain (9.7%) and enlarged kidneys were documented, revealing the profound and distressing extent of the internal afflictions suffered by these fish (Figure 4).

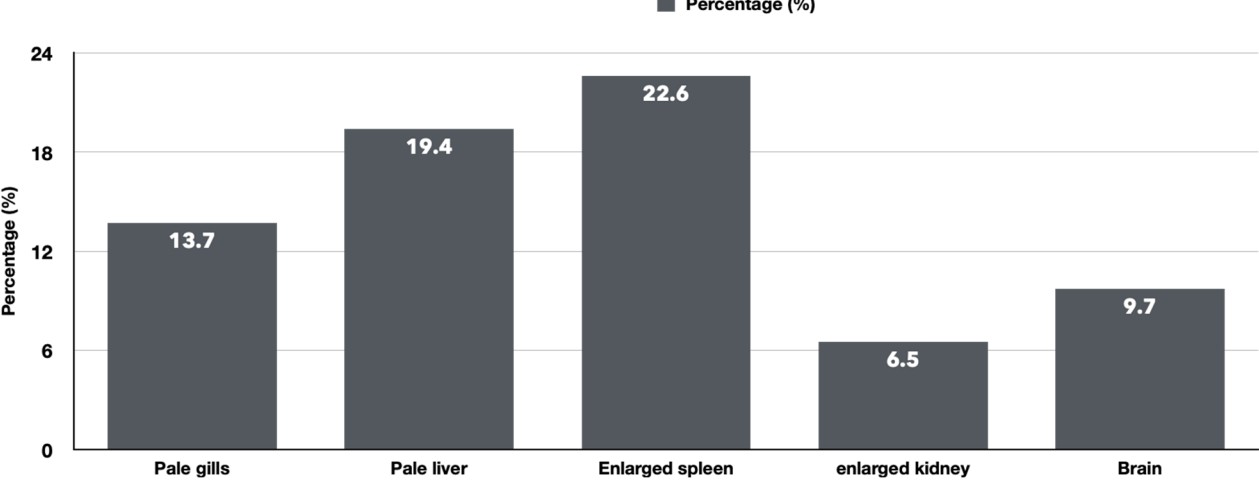

**Figure 4.** The percentage of fish with internal pathological lesions recorded in the moribund fish.

### 3.2. Bacterial Identification

The survey samples yielded a total of 300 bacterial isolates, belonging to 10 genera of bacteria (Table 2). Moribund fish from various farms harboured a diverse range of bacterial species, often coexisting with two to five other types of bacteria. The coexistence of multiple bacterial species within the liver of the diseased fish was consistently observed, with many species simultaneously cohabiting with up to three distinct bacterial genera. Therefore, the liver is a strategic organ that should be used for the diagnosis of bacterial co-infections in diseased tilapia.

**Table 2.** Occurrence of the different bacteria genera from the 11 farms in the study area.

| Bacterial Isolates | Number (%) of Isolated Bacterial Species by Source | | | | | | | | | | | |
|---|---|---|---|---|---|---|---|---|---|---|---|---|
| | Farm 1 | Farm 2 | Farm 3 | Farm 4 | Farm 5 | Farm 6 | Farm 7 | Farm 8 | Farm 9 | Farm 10 | Farm 11 | Total |
| *Acinetobacter* spp. | 0 (0.0) | 1 (0.3) | 1 (0.3) | 4 (1.3) | 3 (1.0) | 0 (0.0) | 2 (0.7) | 3 (1.0) | 2 (0.7) | 3 (1.0) | 2 (0.7) | 21 (7.0) |
| *Aeromonas* spp. | 2 (0.7) | 2 (0.7) | 6 (2.0) | 3 (1.0) | 7 (2.3) | 1 (0.3) | 9 (3.0) | 3 (1.0) | 1 (0.3) | 1 (0.3) | 4 (1.3) | 39 (13.0) |
| *Bacillus* spp. | 2 (0.7) | 2 (0.7) | 3 (1.0) | 1 (0.3) | 1 (0.3) | 1 (0.3) | 2 (0.7) | 1 (0.3) | 1 (0.3) | 0 (0.0) | 0 (0.0) | 14 (4.7) |
| *Citrobacter* spp. | 0 (0.0) | 0 (0.0) | 1 (0.3) | 1 (0.3) | 0 (0.0) | 0 (0.0) | 0 (0.0) | 0 (0.0) | 0 (0.0) | 0 (0.0) | 0 (0.0) | 2 (0.7) |
| *Klebsiella* spp. | 4 (1.3) | 1 (0.3) | 2 (0.7) | 1 (0.3) | 2 (0.7) | 4 (1.3) | 6 (2.0) | 3 (1.0) | 2 (0.7) | 1 (0.3) | 0 (0.0) | 26 (8.7) |
| *Lactococcus* spp. | 6 (2.0) | 1 (0.3) | 3 (1.0) | 0 (0.0) | 2 (0.7) | 2 (0.7) | 4 (1.3) | 2 (0.7) | 1 (0.3) | 1 (0.3) | 0 (0.0) | 22 (7.3) |
| *Micrococcus* spp. | 4 (1.3) | 2 (0.7) | 6 (2.0) | 1 (0.3) | 4 (1.3) | 1 (0.3) | 6 (2.0) | 2 (0.7) | 2 (0.7) | 0 (0.0) | 1 (0.3) | 29 (9.7) |
| *Pseudomonas* spp. | 2 (0.7) | 1 (0.3) | 4 (1.3) | 1 (0.3) | 5 (1.7) | 2 (0.7) | 9 (3.0) | 2 (0.7) | 1 (0.3) | 1 (0.3) | 3 (1.0) | 31 (10.3) |
| *Staphylococcus* spp. | 0 (0.0) | 2 (0.7) | 1 (0.3) | 0 (0.0) | 2 (0.7) | 0 (0.0) | 0 (0.0) | 0 (0.0) | 0 (0.0) | 0 (0.0) | 0 (0.0) | 5 (1.7) |
| *Streptococcus* spp. | 3 (1.0) | 2 (0.7) | 2 (0.7) | 1 (0.3) | 1 (0.3) | 4 (1.3) | 4 (1.3) | 2 (0.7) | 2 (0.7) | 0 (0.0) | 0 (0.0) | 21 (7.0) |
| Unidentified | 2 (0.7) | 3 (1.0) | 3 (1.0) | 2 (0.7) | 5 (1.7) | 2 (0.7) | 7 (2.3) | 2 (0.7) | 1 (0.3) | 1 (0.3) | 2 (0.7) | 30 (10.0) |
| No Growth | 5 (1.7) | 3 (1.0) | 8 (2.7) | 5 (1.7) | 13 (4.3) | 3 (1.0) | 11 (3.7) | 5 (1.7) | 2 (0.7) | 2 (0.7) | 3 (1.0) | 60 (20.0) |
| Total | 30 (10.0) | 20 (6.7) | 40 (13.3) | 20 (6.7) | 45 (15.0) | 20 (6.7) | 60 (20.0) | 25 (8.3) | 15 (5.0) | 10 (3.3) | 15 (5.0) | 300 (100.0) |

The most predominant and prevalent genus of bacteria identified was *Aeromonas* spp. (39/300; 13%). Among the bacterial genera, others that were isolated included *Acinetobacter* spp. (7%), *Klebsiella* spp. (8.7%), *Lactococcus* spp. (7.3%), *Micrococcus* spp. (9.7%), *Pseudomonas* spp. (10.3%), and *Streptococcus* spp. (7.0%; Table 2).

### 3.3. Antibiotic Susceptibility

The results of the susceptibility testing for the six tested antibiotics on the five representative samples of each genus of the 300 bacterial isolates are tabulated in Figure 5.

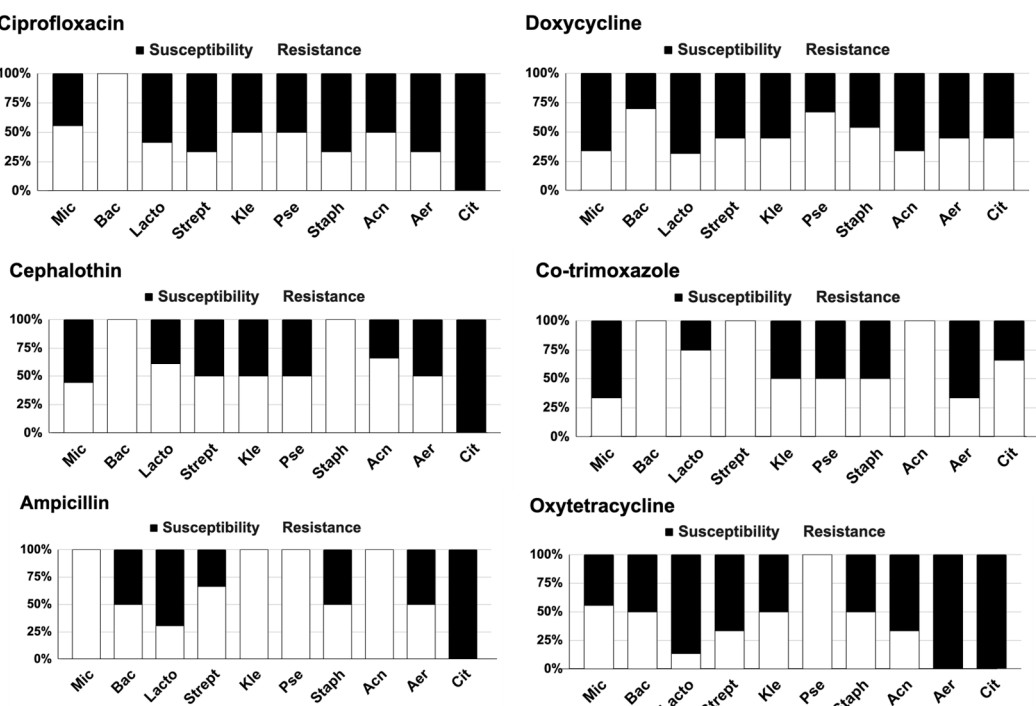

**Figure 5.** The results of antibiotic susceptibility tests performed on the representative isolated bacteria of diseased fish using Ciprofloxacin (30 μg/disc), Doxycycline (30 μg/disc), Cephalothin (30 μg/disc), Co-Trimoxazole (25 μg/disc), Ampicillin (25 μg/disc), and Oxytetracycline (30 μg/disc). Mic = *Micrococcus*, Bac = *Bacillus*, Lacto = *Lactococcus* spp., Strepto = *Streptococcus* sp., Kle = *Klebsiella* spp., Pse = *Pseudomonas* spp., Staphy = *Staphylococcus* spp., Acn = *Acinetobacter*, Aer = *Aeromonas* spp., Cit = *Citrobacter* spp.

The overall susceptibility rates for each antibiotic for the representative bacterial isolates were the highest in doxycycline (52.9%), oxytetracycline (51.7%), and ciprofloxacin (50.3%). Complete resistance to ciprofloxacin, co-trimoxazole, and cephalothin was recorded in the *Bacillus* spp. Doxycycline was the only antibiotic that showed effectiveness against all the bacterial isolates, while other antibiotics were ineffective for one or more bacterial isolates.

## 4. Discussion

With the escalation of tilapia aquaculture production, transforming it into a pivotal source of animal protein in Zambia, it has become imperative to establish stringent surveillance measures and to fortify fish farms against the pernicious onslaught of infectious agents and the insidious threat of emerging or production-related diseases.

In the present study, diseased tilapia exhibited clinical signs such as erratic swimming patterns, as well as lesions including skin discoloration, ulceration, scale loss, fin erosion, and corneal opacity. Notably, macroscopic examination of the internal organs revealed the presence of pale gills, enlarged spleens, congested brains, pallid hepatomegaly, and nephromegaly. Gills, being vital organs directly exposed to the external environment, serve as crucial indicators of various diseases such as parasitosis, bacteriosis, and viral infections, as their visible changes provide valuable insights [19,20]. Furthermore, changes in the appearance of gills serve as an indicator of water quality, which reflects the possibility of acidic or alkaline pH levels or intoxication due to high concentrations of ammonia or nitrite [21]. The erosion of fins serves as an indicator of various conditions, which may include inappropriate handling practices and cannibalistic behaviours in fish within a pond or cage [22]. Consequently, fin damage impacts the growth, survival, and swimming abilities of the fish—thereby affecting their welfare [23]. It should be highlighted that bacterial or fungal infections may exacerbate fin erosion to chronicity, having developed acutely from crowding during activities such as capture, feeding, and handling procedures [23]. A previous study conducted in Zambia reported farmers stocking fish at densities higher than those recommended for pond (>8 fish/m$^2$) and cage (>100 fish/m$^3$) production [14]. Therefore, in the present study, the observation of fish with fin erosion lesions coupled with damaged skin and pale gills would be attributed to the possibility of fish being overstocked in the cages, thereby leading to overcrowding and consequently predisposing them to secondary bacterial infections. It should be highlighted that damage to the skin through the loss of scales and formation of ulcers increases the incidence of exposure to bacterial infections, as the skin which serves as the primary defence against environmental bacteria is compromised [24,25].

The identified bacterial isolates from diseased tilapia reported in the present study provide important baseline information on the pathogens affecting fish in small-holder fish farms on Lake Kariba. This study has identified ten different genera of bacteria from diseased Nile tilapia; the isolates are well-known primary and secondary etiological pathogens that cause disease in tilapia.

*Aeromonas* spp., which was the predominant bacteria isolated from the diseased tilapia in this study, is the causative agent of motile aeromonad septicaemia—a common disease of farmed tilapia. *Aeromonas* spp. are ubiquitous in freshwater aquatic environments and are opportunistic pathogens, only causing disease when fish are stressed in a farming system [9,19]. In other studies, *Aeromonas hydrophila* and *A. veronii* have been reported as the main pathogens in tilapia cultures [26]. Co-infection of Aeromonas with other bacteria is a significant contributor to high mortality rates in tilapia farming [26–28]. In the present study, *Aeromonas* spp. was isolated with *Streptococcus* spp. and *Pseudomonas* spp. from moribund fish, although the determination of co-infection was not the scope of this study. *Aeromonas* spp. and *Streptococcus* spp. co-infection causes significant mortality, and the two bacterial pathogens were isolated from fish in the present study [29].

*Pseudomonas* spp. was the second most prevalent bacteria isolated in the present study, and has been reported to cause Pseudomonas septicaemia [9]. Most *Pseudomonas* spp. are non-pathogenic, but some cause diseases in fish such as Japanese and European eel

and tilapia [30]. *Pseudomonas* spp. is described as an opportunistic pathogen of tilapia experiencing stressful environmental and husbandry circumstances [9,31]. Therefore, in the present study, the infection of tilapia with *Pseudomonas* spp. could be attributed to poor husbandry practices such as high stocking densities, as also indicated by other external lesions such as fin erosion and skin damage. In other studies, they reported reddening of the whole body, abdominal swelling, cloudiness of eyes, loosening scales, and congested gills as the main clinical signs of pseudomonad septicaemia—some which were reported in the present study [10,30].

*Lactococcus* spp. was among the most significant pathogens causing disease in tilapia on small-scale fish farms in the present study. Two species—*Lactococcus garvieae* and *L. petauri*—have been reported to cause mortality in aquaculture [11]. Previous studies have demonstrated the presence of *L. garvieae* in commercial fish farms on Lake Kariba, Zambia [15]. From the present study, *Lactococcus* spp. were isolated from fish with corneal opacity and skin defects. In an infection experiment, Bwalya et al. (2020) reported that *L. garvieae* isolated from Lake Kariba was non-invasive, and therefore required a break in the skin to facilitate the invasion of the bacteria and predispose the fish to lactococcosis [32]. Therefore, as seen in the present study, tilapia with skin ulcers and missing scales are highly predisposed to invasion with *L. garvieae*.

*Streptococcus* spp. was also isolated in this present study. Outbreaks of streptococcosis have been reported in farmed tilapia across the world—mainly caused by *S. agalactiae* and *S. iniae* [11]. In a previous study conducted in Zambia, *S. agalactiae* was isolated from diseased tilapia from large commercial farms on Lake Kariba [15]. In that study, some of the clinical manifestations of the streptococcosis included erratic swimming, corneal opacity, distended abdomen, and haemorrhages, which were also recorded in the present study [15].

In the present study, the isolation of bacteria considered to be non-pathogenic to fish such as *Acinetobacter* spp., *Bacillus* spp., *Citrobacter* spp., *Klebsiella* spp., *Micrococcus* spp., *Staphylococcus* spp., and unidentified bacteria is an indication of immunosuppression in the fish [33]. These bacteria are ubiquitous in freshwater aquatic environments and therefore only gain entry into fish after their immunity is compromised [24]. A previous study conducted in Zambia linked high stocking densities and poor water management practices as risk factors for immunosuppression in farmed fish, which may predispose them to ubiquitous bacteria in the aquatic environment [14].

The results of antibiotic susceptibility tests in the present study showed that all the tested bacterial isolates were sensitive to doxycycline. Therefore, this antibiotic may serve as the best candidate for the treatment of bacterial infections in farmed tilapia among small-scale producers on Lake Kariba. In the present study, ciprofloxacin and oxytetracycline may also represent possible treatment options, as their resistance was only reported in *Bacillus* spp. and *Pseudomonas* spp., respectively. Despite the detection of antibiotic-resistant bacteria in diseased fish, it should be highlighted that antibiotics are not currently used for treating disease outbreaks on Lake Kariba. A study conducted by Ndashe et al. (2023) reported that tilapia farmers used salt (sodium chloride) as the first-line remedy when they encountered any form of disease or observed mortalities [14]. Therefore, the detection of antibiotic-resistant bacteria in diseased farmed tilapia on Lake Kariba raises concerns as to other possible sources of resistance. Other studies have demonstrated that the aquatic environment may contain terrestrial animal bacterial pathogens, which act as agents in sharing genetic determinants between aquatic and terrestrial bacteria [13]. Therefore, the antibiotic-resistant bacteria emerging from human and livestock activities around the lake could be a possible source of resistance genes in aquatic bacteria, causing outbreaks in the fish farms on Lake Kariba. Studies have shown that aquatic environments with aquaculture facilities (cages and nets) provide conditions that allow for horizontal gene transfer [13].

## 5. Conclusions

This study identified bacteria infecting Nile tilapia farmed by small-scale producers and determined the antibiotic susceptibility of selected bacteria. This information is important in the development of fish disease management strategies for farmed tilapia on Lake Kariba. To control acute disease outbreaks, farmers should be encouraged to implement best management practices such as improved handling and compliance to recommended stocking densities. Regarding antimicrobial resistant bacterial strains in aquatic environments, research should be undertaken to understand the source of this resistance.

**Author Contributions:** Conceptualization, K.N. and M.S.; methodology, M.S. and K.N.; software, K.N.; validation, F.C.Z.; formal analysis, C.C., M.S. and K.N.; investigation, M.S., F.C.Z., C.C., M.M.S., K.C., L.M., E.S.K. and B.M.H.; data curation, B.M.H. and K.N.; writing—original draft preparation, K.N. and M.S.; writing—review and editing, K.N., B.M.H., K.C., S.R. and M.M.S.; visualization, K.N.; supervision, B.M.H., M.M.S., K.C. and M.S.; funding acquisition, B.M.H. and S.R. All authors have read and agreed to the published version of the manuscript.

**Funding:** This research was funded by The Feed the Future Innovation Lab for Fish managed by Mississippi State University through an award from USAID (Award No. 7200AA18CA00030; M. Lawrence, PI), which provided support to this project.

**Institutional Review Board Statement:** This study was conducted in accordance with the Declaration of Helsinki and approved by the Institutional Review Board (or Ethics Committee) of ERES Converge (reference number: 2019-AUG-024).

**Informed Consent Statement:** The fish farmers provided informed consent before sample collection. Participants were assured of confidentiality and anonymity regarding the given information.

**Data Availability Statement:** Not applicable.

**Acknowledgments:** This work is made possible by the generous support of the American people provided by the Feed the Future Innovation Lab for Fish through the United States Agency for International Development (USAID). The contents are the responsibility of the authors and do not necessarily reflect the views of USAID or the United States Government. The Feed the Future Innovation Lab for Fish is managed by Mississippi State University through an award from USAID (Award No. 7200AA18CA00030; M. Lawrence, PI), which provided support to this project (B. Hang'ombe, PI; S. Reichley, PI). Furthermore, the authors would like to thank the farmers that participated in the study and colleagues from the Ministry of Fisheries and Livestock, Department of Fisheries of the Republic of Zambia, who facilitated the interaction.

**Conflicts of Interest:** The authors declare no conflict of interest.

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
