# Peer review of "An Investigation of Bacterial Pathogens Associated with Diseased Nile Tilapia in Small-Scale Cage Culture Farms on Lake Kariba, Siavonga, Zambia"

_fishes, doi:10.3390/fishes8090452_

Round 1
Reviewer 1 Report
The manuscript includes the information about bacterial pathogens isolated from Nile tilapia, the one of the world's most important food fishes.
Comments:
Materials and methods:
Examination of samples from recent outbreak is very valuable, but we must ensure that other pathogens have not been detected if authors “investigated bacterial disease outbreaks” . If other studies have been done, the authors should add methods which the presence of other pathogens like viral were excluded. If other studies have not been conducted, it should be clarified.
Line 87: The water temperature and fish weight should be add.
Line 99: The different organs (brain, liver, kidney) were collected from fish. Have all organs taken from each fish? Is should be clarified.
Results:
Line 167: Which bacteria were grown from different organs.
Author Response
[General Comment]: The manuscript includes the information about bacterial pathogens isolated from Nile tilapia, the one of the world's most important food fishes
Authors response: Thank you for extensive review of the manuscript.
Reviewer comment: Examination of samples from recent outbreak is very valuable, but we must ensure that other pathogens have not been detected if authors “investigated bacterial disease outbreaks”. If other studies have been done, the authors should add methods which the presence of other pathogens like viral were excluded. If other studies have not been conducted, it should be clarified.
Authors response: As shown in the methodology our study only focused on bacterial pathogens and did not endeavor to detect viruses.
Reviewer comment: Line 87-88: The water temperature and fish weight should be added.
Authors response: Added text [The water temperature during the study period was 27 to 30.2 °C.]
Reviewer comment: Line 99: The different organs (brain, liver, kidney) were collected from fish. Have all organs taken from each fish? Is should be clarified.
Authors response: Added text [From each moribund fish, the brain, liver, and kidney were sampled using sterile disposable inoculation loops and streaked on MacConkey agar (HiMedia, India), nutrient agar (HiMedia, India), and blood agar (HiMedia, India) using a strictly aseptic technique]
Reviewer comment: Line 167: Which bacteria were grown from different organs.
Authors response: Added text [The coexistence of multiple bacterial species within the liver of the diseased fish was consistently observed, with many species simultaneously cohabiting with up to 3 distinct bacterial genera (data not shown)].

Reviewer 2 Report
A good survey illustrating the bacterial infections experienced in small tilapia farms in Zambia. The unit of interest was clearly presented in the materials an methods and the paper was fairly easy to follow.
Some specific comments and questions
Ln 88 - you indicate moribund fish - are these fish that were swimming on the surface of the cages or brought up from the bottom of the cages along with mortalities.
Ln90 - clove powder please confirm dosage - it appears to be low.
Ln 103- you mention room temperature (what would that be - approximately)
Section2.6 - you list a number of antibiotics examined for sensitivity testing - why were these antibiotics selected? (either in M&M or in Discussion)
Results
Ln 134 - you indicate that 300 fish were sampled on 11 farms - could you provide a range and a mean(or median) here - I do see it is provided in Table 2 but it may be worthwhile here too.
Ln 137 - Necropsy should be necropsy
Section 3.2 - Is there any reason why the bacterial identification did not precede to the species level? Also were any tissues better for culturing the bacterial - it may be helpful for the diagnosticians to know that kidney culture would be sufficient to culture - or that liver and brain together were very good at isolating. Some description would be useful.
Discussion -
Ln 205 - you mention inadequate pH - what do you mean exactly - too acidic? please elaborate.
Ln 213 - you mention high densities - please elaborate.
Author Response
[General Comment]: A good survey illustrating the bacterial infections experienced in small tilapia farms in Zambia. The unit of interest was clearly presented in the materials an methods and the paper was fairly easy to follow.
Authors response: Thank you for extensive review of the manuscript.
Reviewer comment: Line 88 - you indicate moribund fish - are these fish that were swimming on the surface of the cages or brought up from the bottom of the cages along with mortalities.
Authors response: Added text [The moribund fish swimming at the surface were collected from affected cages that reported high mortality within 5 days preceding the farm visit.]
Reviewer comment: Line 90 - clove powder please confirm dosage - it appears to be low.
Authors response: revised text [The fish were humanely euthanized using clove powder at 250 mg/L of water.]
Reviewer comment: Ln 103- you mention room temperature (what would that be - approximately).
Authors response: revised text [The inoculated petri dishes were stored at room temperature (approximately 28 °C)and transported to the bacteriology laboratory, School of Veterinary Medicine, the University of Zambia within 24hrs of sampling.]
Reviewer comment: Section2.6 - you list a number of antibiotics examined for sensitivity testing - why were these antibiotics selected? (either in M&M or in Discussion).
Authors response: added text [The antimicrobial agents were selected as representatives of the different classes of antibiotics used in aquaculture practice.]
Reviewer comment: Line 134 - you indicate that 300 fish were sampled on 11 farms - could you provide a range and a mean(or median) here - I do see it is provided in Table 2 but it may be worthwhile here too.
Authors response: Revised text [A total of 300 samples were collected from 11 farms (27 ± 15 samples per farm) in the duration of the study.]
Reviewer comment: Ln 137 - Necropsy should be necropsy
Authors response: Revised text [The clinical and necropsy findings]
Reviewer comment: Section 3.2 - Is there any reason why the bacterial identification did not precede to the species level? Also were any tissues better for culturing the bacterial - it may be helpful for the diagnosticians to know that kidney culture would be sufficient to culture - or that liver and brain together were very good at isolating. Some description would be useful.
Authors response: Revised text [The coexistence of multiple bacterial species within the liver of the diseased fish was consistently observed, with many species simultaneously cohabiting with up to 3 distinct bacterial genera (data not shown). Therefore, the liver is a strategic organ that should be used for the diagnosis of bacterial co-infections in diseased tilapia.]
The bacterial identification could not precede to species level due to inadequate availability of some reagents on the Zambian market at the time of sample analysis.
Reviewer comment: Line 205 - you mention inadequate pH - what do you mean exactly - too acidic? please elaborate.
Authors response: Revised text [Furthermore, changes in the appearance of gills serve as an indicator of water quality, which reflects the possibility of acidic or alkaline pH levels or intoxication due to high concentrations of ammonia, or nitrite [21].]
Reviewer comment: Line 213 - you mention high densities - please elaborate.
Authors response: Revised text [A previous study conducted in Zambia reported farmers stocking fish at densities higher than the recommended for pond (>8fish/m2) and cage (>100fish/m3) production [14]]
